# Pregnancy rest-activity patterns are related to salivary cortisol rhythms and maternal-fetal health indicators in women from a disadvantaged population

**Theresa Casey**[1]*, **Hui Sun**[2], **Aridany Suarez-Trujillo**[1], **Jennifer Crodian**[1], **Lingsong Zhang**[2,3], **Karen Plaut**[1], **Helen J. Burgess**[4], **Shelley Dowden**[5], **David M. Haas**[5], **Azza Ahmed**[6]

**1** Department of Animal Sciences, Purdue University, West Lafayette, IN, United States of America, **2** Department of Statistics, Purdue University, West Lafayette, IN, United States of America, **3** Regenstrief Center for Healthcare Engineering, Purdue University, West Lafayette, IN, United States of America, **4** Department of Psychiatry, University of Michigan, Ann Arbor, MI, United States of America, **5** Department of Obstetrics and Gynecology, Indiana University, Indianapolis, IN, United States of America, **6** School of Nursing, Purdue University, West Lafayette, IN, United States of America

* theresa-casey@purdue.edu

**Data Availability Statement:** All relevant data are within the manuscript and its Supporting Information files.

## Abstract

Irregular rest-activity patterns can disrupt metabolic and hormonal physiology and potentially lead to disease. Little is known regarding rest-activity patterns during gestation and their association with hormonal rhythms and health in pregnant women. We conducted a pilot study to determine if 24 h rest-activity was related to saliva cortisol rhythms and maternal-fetal health in an economically disadvantaged population. Primiparous women wore a wrist actigraphy device for a week to record activity during gestational weeks 22 (G22; $n = 50$) and 32 (G32; $n = 46$) and postpartum week one (PPW1; n = 39). Participants collected saliva samples every 4 hr over a 24 hr period during G22 (n = 22), G32 (n = 20) and 24–48 hr postnatal (n = 20), and cortisol concentrations were measured with ELISA. Circadian rhythmicity was assessed using autocorrelation coefficient ($r24$) and cosinor analysis. Blood glucose levels, body mass index (BMI), gestational disease data, and gestational age of infant at birth were abstracted from medical charts. Time of cortisol peak (acrophase) during G22 was related with acrophase of activity ($r = 0.66$; $p = 0.001$) and blood glucose levels ($r = 0.58$; $p = 0.006$). During G22, minutes of wake after sleep onset was positively related to cortisol mesor and AUC ($p < 0.05$). Rest-activity $r24$, $R^2$, and mesor during G32 were positively ($p < 0.05$) associated with gestational age of infant at birth. Across all three time points $r24$ of activity was related with cortisol amplitude ($r = 0.33$; $p = 0.01$). Findings support a relationship between rest-activity patterns and saliva cortisol rhythms during pregnancy. The association of less robust activity rhythms with earlier gestational age of infant at birth indicates a potential link between circadian system disruption and maternal-fetal health outcomes.

**Funding:** This project was funded by Indiana AgSeed; the funders had no role in study design, data collection and analysis, decision to publish, or preparation of the manuscript.

**Competing interests:** The authors have declared that no competing interests exist.

## Introduction

Physiology and behavior of women change extensively throughout pregnancy and around the time of parturition. Relative to other physiological states in women's lives, the day-to-day stability of rest-activity rhythms during late gestation and the early postnatal period is poor and diminishes throughout pregnancy into the early postpartum period [1, 2]. Rest-activity patterns are connected with circadian structure. In non-pregnant women irregular rest-activity patterns associated with shift work, psychological stress and irregular sleep patterns reflect circadian disruption and affect both physiological and neurobehavioral health [3]. Although we know that metabolic, circadian, and reproductive systems are integrated and reciprocally regulated [4–7], very little is known regarding patterns of rest-activity during gestation and their relationship to hormonal and metabolic adaptations to pregnancy.

Understanding the relationship of rest-activity rhythms with hormones and metabolism is important as growing evidence suggest that circadian disruption associated with irregular rest-activity patterns during pregnancy negatively impacts maternal and fetal health. Retrospective analysis of factory workers found lighter birth weight babies significantly associated with maternal night-shift work [8]. Moreover, a population-based prospective study identified maternal sleep deprivation ($\leq 8h$) and shift work as risk factors for small for gestational age infants [9]. Working consecutive night shifts and quick returns after night shifts during the third trimester was found associated with an increased risk of hypertension, particularly among obese women [10]. Furthermore, meta-analysis found shiftwork increased relative risk of preterm delivery, low birthweight, and small-for-gestational-age infants [11].

The circadian rhythm of circulating cortisol is a primary output of the central clock and functions to synchronize the timing of physiological processes across the body [12, 13]. Circadian rhythms of saliva cortisol, which reflects free-bioavailable cortisol, are maintained throughout pregnancy, despite a progressive decline in the cortisol awakening response and maternal stress reactivity [14, 15]. Studies on the relationship between cortisol circadian rhythms and maternal health outcomes are limited, but indicate that they may be related. For example, studies of waking and evening salivary cortisol in early pregnancy (median gestational week 14) and late pregnancy (median gestational week 30) found women who experienced a stressful life event or were concerned about pregnancy complications during the second trimester had 27% higher evening cortisol. Morning cortisol levels were unaffected [14]. Shorter gestation length was associated with higher salivary cortisol concentrations at awakening and throughout the day; with women delivering at 36 weeks gestations having 13% higher cortisol levels at awakening than women delivering at 41 weeks gestation [16]. Women with prepregnancy obesity had higher evening cortisol at 35 weeks of gestation compared to women who were not obese prior to pregnancy, however there was no significant association among prepregnancy obesity and cortisol at 24 weeks [17].

We hypothesized that irregular rest-activity patterns during pregnancy and the peripartum alter cortisol circadian rhythms and potentially affect maternal-fetal health. Our first step to testing this hypothesis was to conduct a pilot study to determine if 24 h rest-activity and salivary cortisol rhythms in pregnant and early postnatal women were related during gestational weeks 22 and 32 and the first week postpartum. Secondly, we analyzed whether there was a relationship between rest-activity rhythms and cortisol rhythms with maternal-fetal health indicators including prepregnancy BMI, development of gestational related disease, blood glucose levels, gestational age of infant at birth, mood and sleep symptoms as well as actigraphic sleep measures in a population of economically disadvantaged women.

## Methods and materials

### Population and setting

A longitudinal observational prospective cohort study was approved by Institutional Review Boards (IRB) at Purdue University, Indiana University, and Eskenazi Health (#1405014855) and was conducted from August 2014 to October 2015. All the women recruited were primiparous, 18–40 years of age, expecting a singleton infant, less than 22 weeks gestational age, and willing to feed baby breast milk. A convenience sample of 92 women was enrolled, and 50 women completed gestational week 22, 46 completed gestational week 32, and 42 women completed all aspects of the study. Loss of fetus, moving or other life events were reasons for half of study withdrawals. The remaining were withdrawn by research staff due to lack of compliance to study protocol. There was no difference in demographic variables between recruited, retained and withdrawn populations [18, 19].

### Data collected and study time line (S1 Fig)

**Initial data collection.** During the initial contact with the participants, a survey was used to gather demographic data, following written and verbal consent. In addition to study consent, participants provided written consent to abstract protected health information (PHI) from the individual's medical records and the medical records of her infant. PHI data collected included pre-pregnancy body mass index (BMI) information, blood glucose levels after 50 g glucose challenge test, and diagnosis of hypertension, preeclampsia and gestational diabetes mellitus (GDM) during pregnancy. The 50 g glucose challenge test was performed between 24–28 weeks pregnancy according to The American Congress of Obstetrics and Gynecologists (ACOG) guidelines. Also abstracted from charts was gestational age of infant at time of birth and infant birth weight.

**Actigraphic data collection.** Participants were asked to wear a wrist actigraph device (Actiwatch Spectrum, Philips Respironics, Andover, MA) on their non-dominant wrist and keep a sleep diary for seven consecutive days during gestational weeks 22 and 32 and postpartum week 1 (days 7–14 after birth of infant). Prior to gestational week 22 and 32 a research assistant contacted study participants to schedule time of pick-up actiwatch; at time of pick up instructions on how to wear device and enter sleep log data were given; a time was also arranged for drop-off of device at study site at completion of data collection period. Transportation costs and logistics were covered by the study. Participants were given the actiwatch device at time of hospital discharge; watches were programmed to begin recording beginning at 7 days postpartum. Actigraphic data were collected in 30-sec epochs and Actiware software (Version 5.04; Respironics, Inc.) was used to export activity counts into excel files (https://purr.purdue.edu/publications/3376) for calculation of the autocorrelation coefficient at 24 hr ($r24$) and cosinor analysis. Actigraphic sleep variables: (1) sleep onset time, (2) end sleep time, (3) total sleep time, (4) sleep efficiency, (5) time in minutes of wakefulness after sleep onset during the night (WASO) and fragmentation index (FragI) were calculated as previously described [18].

**Saliva collection.** A subset of women was recruited (n = 38) to measure circadian profiles of saliva cortisol levels at gestation week 22 and 32 and postpartum 24–48 hr. When we reached our targeted number of women (n = 25) for returning samples, this arm of the study was closed. Participants were given saliva sampling kits and instructions on how to sample when they came to pick up the actiwatch for collection of samples at home during gestational week 22 and 32 study periods. For postpartum 24–48 hr period, a research assistant visited each participant in the hospital to give instructions and collect and store vials. For home

collections during pregnancy, participants were asked to begin sampling after the 7 days of actiwatch data collection was completed, and to start sampling from the time of waking up. Participants collected saliva samples every four hours for a 24-hour period (from time of waking to the next awakening), for a total of seven samples. Approximately 24 hours after participants delivered their infant, they were asked again to collect saliva samples every 4 hr beginning from 24 hr to 48 hr after delivery. This regimen was selected to standardize sampling times relative to birth since levels of circulating cortisol are expected to change dynamically around the time of parturition. Written instructions given to participants included requests not to drink or eat for at least 60 minutes before the saliva collection, and to maintain their regular sleep habits, despite waking up to collect saliva samples. Subjects were asked to store samples in their home freezer until they transported them to the study site on an ice pack provided in the kit.

*Salivary cortisol analysis.* Salivary cortisol levels were analyzed using commercial enzyme immunoassay kits (Salimetrics, PA). The inter-assay coefficient of variation (CV) was 10%, and the intra-assay coefficient of variation (CV) was 3%. All samples were run in duplicate and the mean value of the duplicate results used.

*Data and statistical analysis.* Robustness of rest-activity circadian rhythms can be assessed by fitting actigraphy data to a cosine curve using the cosinor method, which calculates the relative fit ($R^2$) as well as mesor (rhythm-adjusted mean), amplitude, and acrophase of the rhythms. The 24-hour autocorrelation coefficient ($r24$) is also a measure of circadian rhythmicity and is calculated by comparing activity data collected during each epoch (a defined period of time, for example 30 sec or 1 min) of a 24-hour period with the activity levels during subsequent 24-hour periods [20]. A strong circadian rhythm is indicated by a good correlation between activity levels during epochs separated by 24 hours. Thus, $r24$ estimates the strength of the circadian periodicity, and in theory ranges between -1 and 1 [21].

Cosinor analysis of actigraphic activity recordings and cortisol levels was performed using the *Cosinor* package in R 3.2, with the assumption that the period is known and is synchronized to 24-hour cycle. The corresponding mesor, amplitude and acrophase were calculated. The AUC was also calculated for cortisol levels for each subject during each sampling period.

The regression model for a single cosinor can then be written as

$$Y(t) = M + A\cos(2\pi t/\tau + \phi) + e(t) \tag{1}$$

where M is mesor (Midline Statistic of Rhythm, a rhythm adjusted mean), A is amplitude (a measure of distance from the midline to the top of the crest or bottom of trough), $\phi$ is the acrophase (a measure of time to where the peak happens), $\tau$ is the period of one cycle, which is set as 24 hours in our study. And the e(t) is the error term which is assumed to be independent and normally distributed.

By assuming $\beta = A\cos(\phi), \gamma = -A\sin(\phi), x = \cos(2\pi t/\tau)$ and $z = \sin(2\pi t/\tau)$, Formula (1) can be easily be rewritten as a linear regression model

$$Y(t) = M + \beta x + \gamma z + e(t) \tag{2}$$

Which can be easily calculated by any statistical package. After estimating the parameters in (2), the amplitude and acrophase in (1) can be calculated by

$$\hat{A} = \sqrt{\hat{\beta}^2 + \hat{\gamma}^2}$$

$$\hat{\phi} = acrtan(-\hat{\gamma}/\hat{\beta}) + K\pi$$

Here K is an integer. The correct value of $\hat{\phi}$ is determined by taking both the sign of $\hat{\gamma}$ and $\hat{\beta}$ into consideration.

All participant data collected were entered and stored on a secured server (RedCap). Statistical analysis was conducted using R; $p$-value≤0.05 was considered significant; a $p$-value >0.05, but <0.1, was discussed as a tendency, or weak relationship. Repeated measures analysis of variance (ANOVA) was used to determine effect of gestational or postpartum time point on autocorrelation coefficient, followed by Bonferoni adjustments for multiple comparisons. Wilcoxon signed-rank test was used to compare activity and cortisol cosinor analysis variables across the three study timepoints. Because of the limited sample size we were not able to determine the relationship of variables in groups of participants with and without diagnosis of preeclampsia, hypertension, or gestational diabetes (GDM) alone, so despite being etiologically distinct diseases, women were categorized as diagnosed with gestational related disease or not [approximately 1/3 of participants; please see [18] and [19]]. The Mann-Whitney U test was used to determine effect of categorical BMI (r < 25 or >25) or diagnosis of gestational related disease (yes or no for diagnosis of hypertension, preeclampsia and, or GDM during pregnancy) on *r24* of activity or rest-activity and saliva cortisol cosine fit calculated variables. Both Pearson and Spearman correlation analysis were used to evaluate the relationship between cortisol cosine analysis variables, activity cosinor variables and *r24* with continuous BMI, gestational age at birth and blood glucose levels from glucose tolerance test. Pearson correlation was used to test the correlation between two normal variables, whereas Spearman analysis dealt with any type of data as it is not sensitive to outliers.

## Results

### Demographics of the study population

The mean age of study participants was 23 ± 3.8 years old, with 64% indicating they were Black or African American and of low socioeconomic status. Detailed descriptions of the sample of the population that completed actigraphy [18] and Pittsburgh Sleep Quality Index (PSQI) and Edinburgh Postnatal Depression Scale (EPDS) surveys [19] are available in our previous publications. The demographics of the population that completed the saliva collection arm was similar to the population that completed actigraphy and PSQI survey components of the study (Table 1). Our aim was to have 25 women complete the saliva arm of our study. Due to quality of samples (for example small amounts) or incorrect sampling protocol (as noted in participant logs) analysis of cortisol levels was conducted on samples from 22, 20 and 20 women at gestation weeks 22, 32 and postpartum period, respectively (Table 1).

### Circadian rhythms of rest-activity during gestational weeks 22 and 32 and postpartum week one

Actigraphic data plots of mean activity at 30-sec epochs across 6 days and all participants show clear circadian rhythms of rest-activity during the three study time points (Fig 1 and S2 Fig). Plots of actigraphic data of an individual participant demonstrates a loss of diurnal pattern of rest-activity in the first week postpartum relative to gestational time points, which was characteristic of multiple women (S2 Fig). A comparison of $R^2$ values calculated from cosine analysis of rest-activity data revealed a significant decrease in fit of the data to a 24-hr rhythm between gestational weeks 22 and 32 and then again between gestational week 32 and postpartum week one (Table 2). The amplitude was significantly decreased between the second and third trimester recordings, and dropped again between gestational week 32 and postpartum week one. The timing of the acrophase (peak) of activity shifted to nearly an hour earlier from a mean time of

**Table 1. Demographic and health characteristics of saliva cortisol study sample** *(n = 24).*

| Characteristics | *n* (%) |
|---|---|
| Race | |
| African American | 15 (63) |
| White | 2 (8) |
| More than one race, or other | 4 (16) |
| Unknown/not reported | 3 (13) |
| Ethnicity | |
| Hispanic | 5 (21) |
| not Hispanic or Latino | 19 (79) |
| Education | |
| Graduate degree | 3 (13) |
| Bachelor degree | 2 (8) |
| Associate degree | 1 (4) |
| High school or GED | 16 (67) |
| No high school or GED | 2 (8) |
| Yearly household income | |
| Less than $10,000 | 12 (50) |
| ≥$10,000, but <$25,000 | 3 (13) |
| ≥$25,000, but <$50,000 | 7 (29) |
| ≥$50,000 | 2 (8) |
| Diagnosis | |
| Pre-eclampsia | 3 (13) |
| GDM | 4 (16) |
| Hypertension | 6 (25) |
| Any gestational disease[a] | 7 (29) |
| BMI >25 | 13 (54) |
| Participants with complete saliva samples | |
| Gestational week 22 | 22 (92) |
| Gestational week 32 | 20 (83) |
| Postpartum 24–48 h | 20 (83) |
| All three time points | 15 (50) |

Data presented describe the demographics and rate of diagnosis among the 26% of participants (*n* = 24) who participated in the saliva arm of the study. (The total sample size was *N* = 92).

[a]Any gestational disease is percent of population with any diagnosis of preeclampsia, hypertension and/or gestational diabetes mellitus (GDM); therefore, percent not additive across diagnosis. BMI = Body mass index.

16:45 at gestational weeks 22 to 15:38 during gestational week 32, and then shifted ahead by more than hour for a mean peak in activity at 16:52 during postpartum week one (Table 2).

During gestational week 22 and 32 and postpartum week one, mean autocorrelation coefficients of activity were 0.14, 0.13, and 0.06, respectively (Fig 2). The *r*24 values were lower than those reported for other populations [21, 22], and previous studies of pregnant women [2]. Actigraphic data were collected in 30 sec epochs for our study, whereas other studies used 1 min epochs. We tested the effect of collapsing data and increasing time intervals to 1 min, 5 min, 10 min and 30 min. As the time interval was increased, the *r*24 values increased. For example, collapsing gestational week 22 data to 1 min intervals increased *r*24 to 0.17, whereas mean *r*24 increased to 0.35 when 30 min intervals were applied. However, the difference between *r*24 during gestational weeks 22 and 32 became less significant. The mean *r*24 using

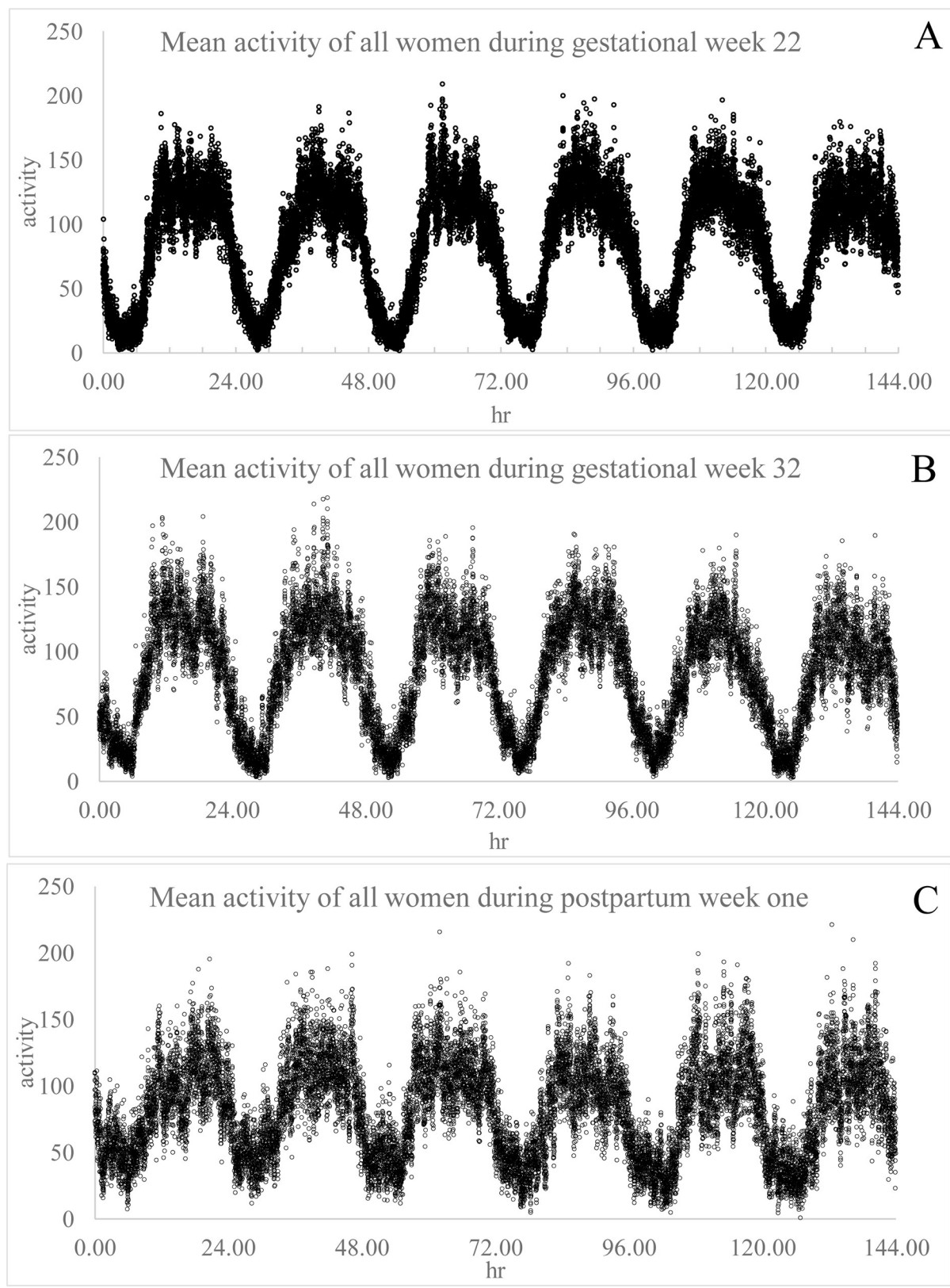

**Fig 1.** Mean activity of women every 30 sec across 6 days of recording during gestational week 22 (A) 32 (B) and postpartum week one (C). Data were averaged among all participants by time point across the first six days of recording.

**Table 2. Cosinor analysis variables of activity rhythms during gestational week 22 (G22), gestational week 32 (G32) and postpartum week 1 (PPW1).**

| | G22 $n = 50$ | G32 $n = 46$ | PPW1 $n = 39$ | p-value of difference | |
| --- | --- | --- | --- | --- | --- |
| | | | | G22 vs. G32 | G32 vs. PPW1 |
| $R^2$ (sd) | 0.14 (0.05) | 0.13 (0.07) | 0.06 (0.04) | 0.35 | 0.17 |
| Mesor (sd) | 84.97 (26.99) | 84.40 (23.83) | 81.92 (26.35) | 0.04 | <0.0001 |
| Amplitude(sd) | 67.58 (25.26) | 62.54 (24.63) | 42.53 (18.23) | 0.03 | <0.001 |
| Acrophase (sd) | 16:45 (3.25) | 15:38 (2.49) | 16:52 (1.75) | 0.007 | <0.0001 |

30 sec epochs for calculations was different between gestational weeks 22 and 32 ($p<0.05$; Fig 2). Mean $r24$ was also significantly different between gestational week 32 and postpartum week one (Fig 2). The difference in $r24$ between gestational weeks 22 and 32 was lost when 1

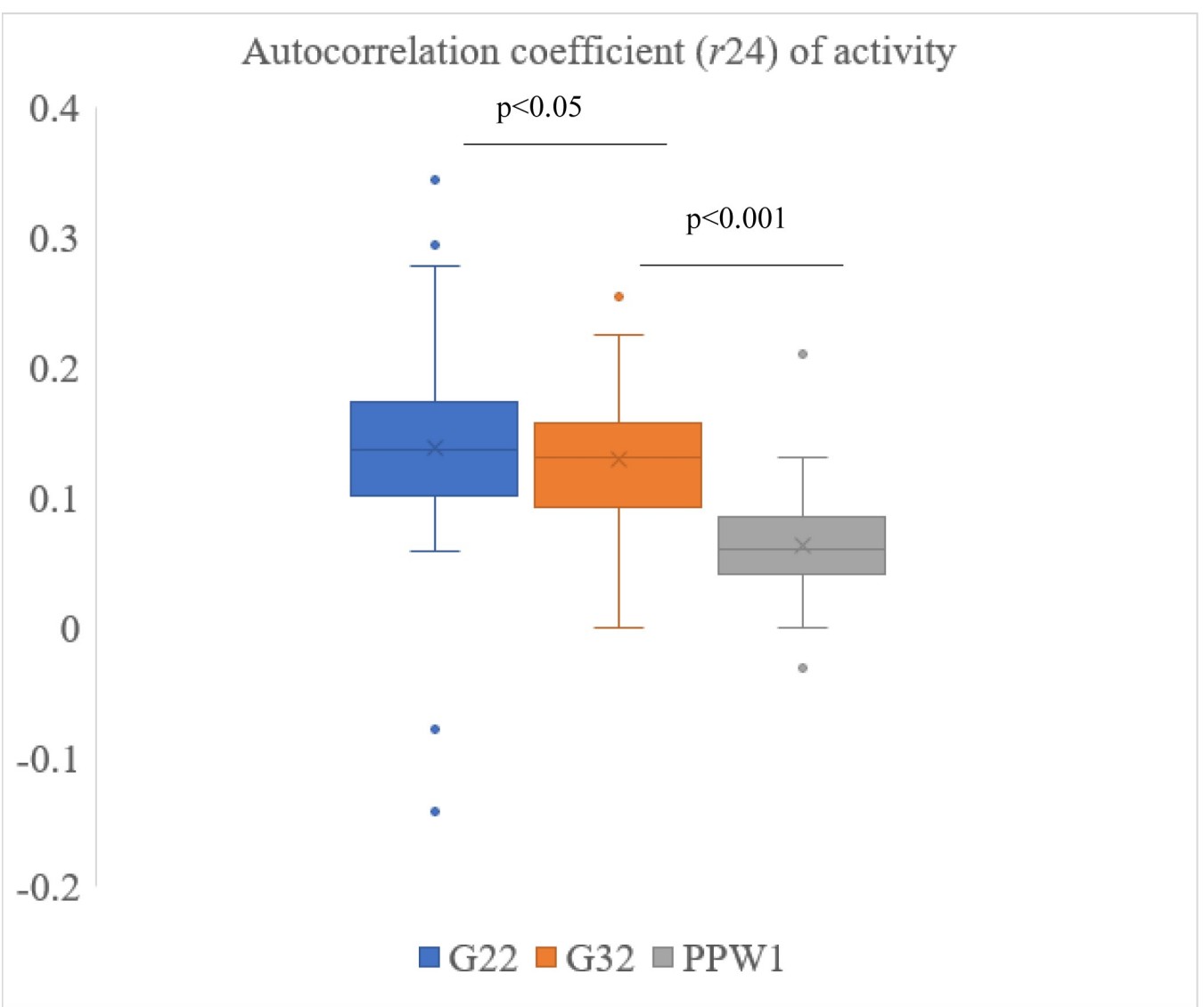

**Fig 2. The 24 h autocorrelation coefficient ($r24$) of rest-activity data during gestational week 22 (G22) and 32 (G32) and postpartum week 1 (PPW1) calculated using 30 sec epochs.** Repeated measures ANOVA was used to determine effect of gestational or postpartum time point on $r24$, followed by Bonferoni adjustments for multiple comparisons.

min intervals of activity were used for calculations. However, the difference between gestational time points and postpartum week one *r*24 remained through 30 min intervals.

## Circadian rhythm analysis of salivary cortisol in women during gestational weeks 22 and 32 and 24–48 h postnatal

Study participants were asked to collect saliva samples every 4 hr over a 24 hr period during gestational weeks 22 and 32 relative to waking to capture circadian rhythms (Fig 3A and 3B; S3A Fig and S3B Fig). Saliva circadian sample collection in the early postpartum was begun relative to the time of parturition, with all samples collected beginning at 24 hr from birth of infant (Fig 3C and S3C Fig). Cosine regression analysis found the acrophase of cortisol was reached 18.34 hr after start of postpartum sampling which was approximately 42 hr from the time of birth. To enable comparison of time of acrophase across all sampling periods, data were expressed relative to clock time (Fig 3D). Cosine fit regression analysis found time of acrophase was not significantly different across the study period, with it calculated as 09:07 during gestational week 22, 09:38 during gestational week 32 and 08:57 for the postnatal 24–48 hr period (Table 3).

The AUC of cortisol concentration was not different between gestational week 22 and 32, but was decreased (*p* = 0.009) between the third trimester sampling period and the first 24–48 hr post-partum (Table 3). Similarly, mesor (or mean cortisol levels across the 24 hr sampling)

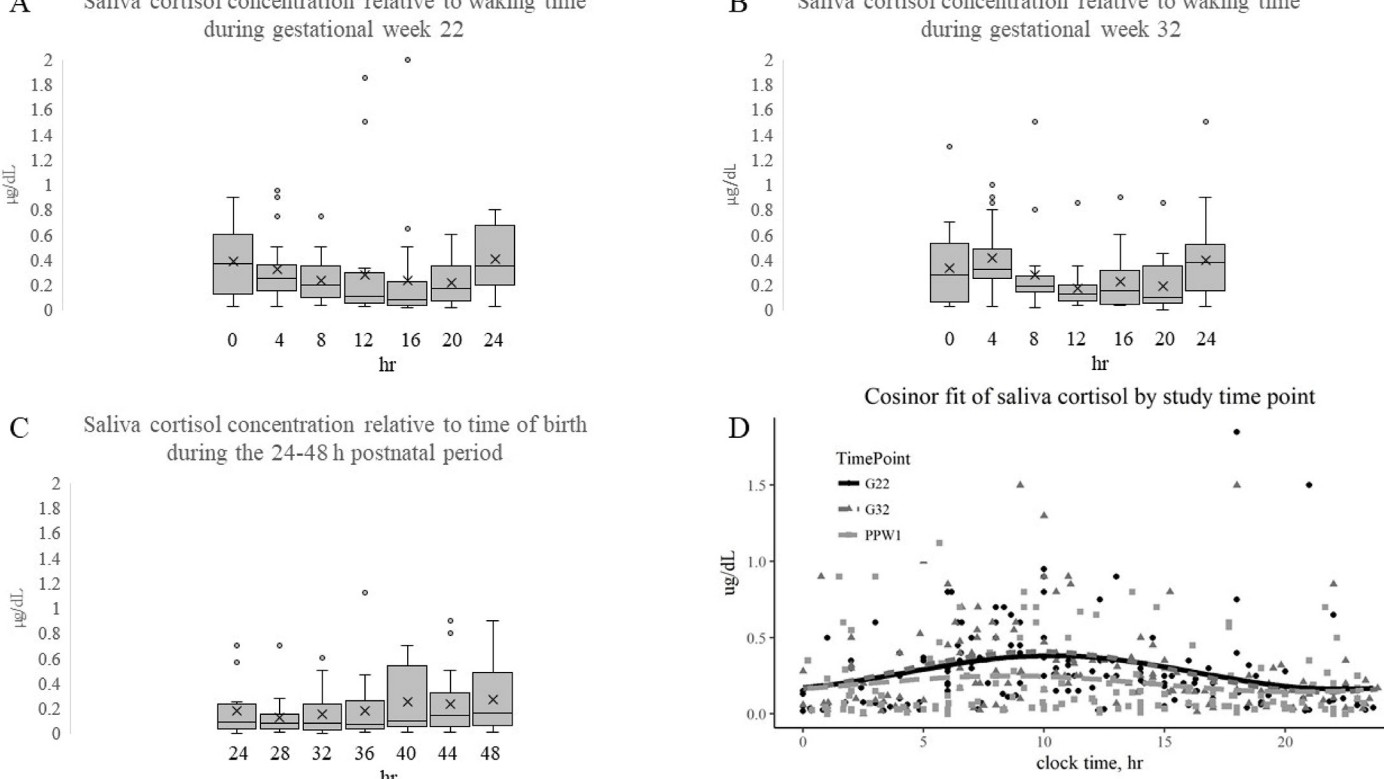

**Fig 3. Saliva cortisol concentrations across a 24 h period during gestational weeks 22 and 32, and 24–48 h postpartum.** Box plots of saliva cortisol relative to time of waking (0 h) during gestational week 22 (A) and 32 (B); or relative to time of birth, beginning at 24 h postpartum. Horizontal line within box plot indicates median, X indicates mean, and dots are outliers. Scatter plot of saliva cortisol levels versus clock time (D), with cosine fitted curve of data collected during gestational week 22 (red), 32 (black) and post-partum 24–48 h (blue).

**Table 3. Mean of cosinor analysis variables of saliva cortisol during gestational week 22 (G22), gestational week 32 (G32) and postpartum 24–48 hr (PP24-48).**

| | G22 (n = 22) | G32 (n = 20) | PP24-48 (n = 20) | p-value of difference | |
| --- | --- | --- | --- | --- | --- |
| | | | | G22 vs. G32 | G32 vs. PPW1 |
| Mesor (sd) µg/dL | 0.29 (0.15) | 0.27 (0.15) | 0.19 (0.13) | NS | <0.05 |
| Amplitude (sd) µg/dL | 0.22 (0.21) | 0.19 (0.11) | 0.12 (0.11) | NS | <0.05 |
| Acrophase (sd) clock time | 09:07 (1.08) | 09:38 (1.16) | 08:57 (1.63) | NS | NS |
| AUC (sd) µg/dL per 24 h | 5.88 (3.90) | 5.77 (3.19) | 3.94 (2.84) | NS | <0.01 |

NS indicates not significant

was not different between gestational week 22 and 32, but dropped ($p<0.05$) between gestational week 32 and the postpartum sampling periods (Table 3). The amplitude of cortisol level also dropped ($p <0.05$) from gestational week 32 to postpartum 24–48 hr, but was not significantly different between gestational week 22 and 32 (Table 3).

## Analysis of the relationship among activity and cortisol rhythms during gestational weeks 22 and 32 and in the postpartum with health outcomes

The relationship of fit of activity rhythms to a 24 hr period ($R^2$), autocorrelation coefficient ($r24$), mesor, amplitude, and acrophase with maternal health indicators was analyzed by comparing variables between groups of women with prepregnancy BMI greater or less than 25, or without and with diagnosis of gestational related disease at each study time point. Women with prepregnancy BMI greater than 25 were found to have lower ($p = 0.01$) mesor of activity during gestational week 32 (Fig 4A) and lower ($p = 0.03$) $r24$ during postpartum week one than women with BMI less than 25 ($r24 = 0.07 \pm 0.03$ and $r24 = 0.04 \pm 0.04$ for BMI <25 and >25, respectively). During gestational week 32, women diagnosed with gestational related disease (hypertension, gestational diabetes mellitus and/or preeclampsia) had lower ($p = 0.014$) amplitudes of activity (Fig 4B) and poorer ($p = 0.021$) cosinor curve fitting compared to participants without disease ($R^2 = 0.15 \pm 0.06$ versus $R^2 = 0.10 \pm 0.06$ for without and with disease, respectively). There was also evidence for a potential relationship ($p = 0.08$) of women with gestational related disease having lower $r24$ during gestational week 32.

There was a positive relationship between gestational age of infant at birth and $r24$ ($r = 0.44$; $p = 0.03$; Fig 4C), $R^2$ ($r = 0.37$; $p = 0.01$) and mesor ($r = 0.38$; $p = 0.01$) of activity during gestational week 32 (Table 4). There was also evidence for a potential relationship between $r24$ at gestational week 22 and gestational age of infant at birth ($r = 0.28$; $p = 0.06$). Prepregnancy BMI (continuous variable) was negatively related to $r24$ ($r = -0.25$; $p = 0.06$; Fig 4D) and mesor (-0.27; $p = 0.07$) of activity during gestational week 32 (Table 4). There was no relationship between activity variables and blood glucose levels.

Women diagnosed with a gestational related disease had significantly ($p = 0.015$) lower mesor of cortisol saliva concentration [0.09 (0.05) µg/dL; mesor (s.d.)] during the postpartum sampling period than those without gestational-related disease [0.22 (0.14) µg/dL]. Similarly, AUC was different ($p<0.05$) between these groups; women diagnosed with gestational related disease [1.8 (1.1) µg/dL per 24 hr] had significantly lower AUC cortisol levels during postpartum sampling period than those who were not [4.65 (2.9) µg/dL per 24 hr]. Interestingly, AUC values for cortisol at gestational week 32 showed the opposite trend, with women diagnosed with gestational related disease [0.30 (0.05) µg/dL per 24 hr] tending ($p = 0.09$) to have higher cortisol levels than those without disease [0.27 (0.17) µg/dL per 24 hr].

Spearman correlation analysis was used to determine if circadian cortisol variables were related to prepregnancy BMI (continuous), blood glucose level and gestational age at birth.

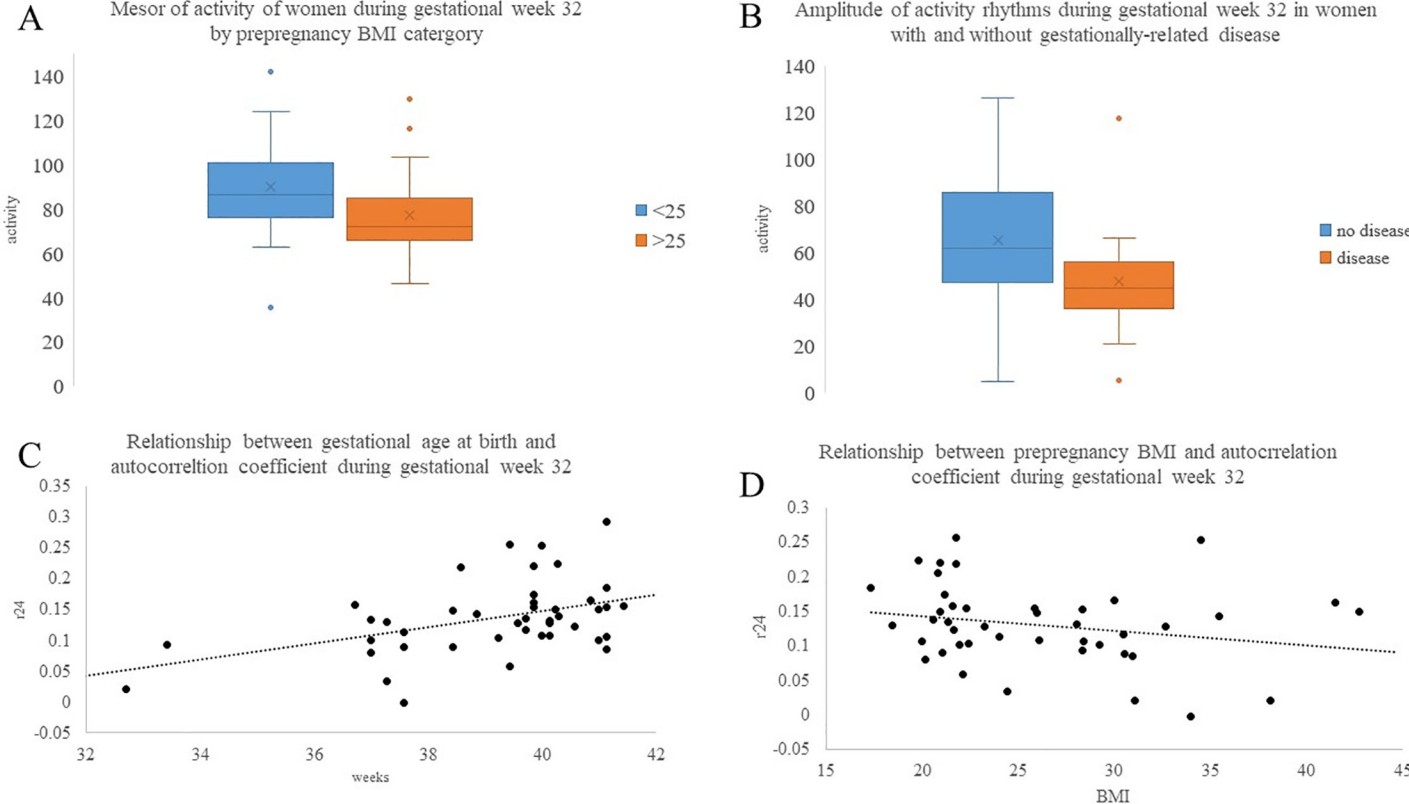

**Fig 4. Relationship of rest-activity circadian rhythm variables to maternal-fetal health indicators.** Mesor of activity during gestational week 32 was different between women with prepregnancy BMI <25 (blue) and >25 (orange) at $p = 0.01$ (A). Amplitude of activity during gestational week 32 was different between women without (orange) and with (blue) diagnosis of gestational related at $p = 0.014$ (B). Horizontal line within box plot indicates median, X indicates mean, and dots are outliers. Spearman correlation analysis of the relationship between gestational age of infant at birth and $r24$ during gestational week 32, $r = 0.44$; $p = 0.03$ (C), and the relationship between prepregnancy BMI (continuous variable) to $r24$ during gestational week 32, $r = -0.25$; $p = 0.06$ (D).

**Table 4. Regression analysis of relationship between maternal-fetal health indicators and actigraphic sleep variables with rest-activity and cortisol rhythm variables by study time point.**

| Variable | Variable | Time point | $r$ | $p$-value |
|---|---|---|---|---|
| Gestational age of infant at birth | Activity $r24$ | G22 | 0.28 | 0.06 |
| | Activity $r24$ | G32 | 0.44 | 0.03 |
| | Activity $R^2$ | G32 | 0.37 | 0.01 |
| | Activity mesor | G32 | 0.38 | 0.01 |
| | Cortisol mesor | PP 24–48 | 0.47 | 0.03 |
| | Cortisol AUC | PP 24–48 | 0.5 | 0.02 |
| BMI | Activity $r24$ | G32 | -0.24 | 0.06 |
| | Activity mesor | G32 | -0.27 | 0.07 |
| | Cortisol mesor | PP 24–48 | -0.47 | 0.03 |
| | Cortisol AUC | PP 24–48 | -0.39 | 0.09 |
| Blood glucose | Cortisol acrophase | G22 | 0.58 | 0.006 |
| | Cortisol acrophase | G32 | 0.48 | 0.07 |
| WASO | Cortisol mesor | G22 | 0.58 | 0.03 |
| | Cortisol AUC | G22 | 0.69 | 0.002 |
| Sleep efficiency | Cortisol AUC | G22 | -0.52 | 0.03 |
| Activity acrophase | Cortisol acrophase | All | 0.66 | 0.001 |

The time of saliva cortisol acrophase was positively related to blood glucose levels during gestational week 22 (Fig 5A); this relationship was somewhat lost by gestational week 32 (Table 4). Postpartum cortisol mesor (Fig 5B) and AUC were negatively related with BMI (Table 4). Mesor and AUC (Fig 5C) during the postpartum sampling period were positively related to gestational age at birth.

Correlation analysis of activity rhythm variables with cortisol variables found the timing of cortisol acrophase was significantly related to the timing of peak activity ($r = 0.66$; $p = 0.001$) during gestational week 22. Moreover, during gestational week 22, minutes of wake after sleep onset (WASO) was positively related to cortisol mesor and AUC, whereas sleep efficiency was negatively related to cortisol AUC during gestational week 22 (Table 4). In the final analysis, data from all three time points were combined to determine the relationship of $r24$, cortisol, self-reported sleep quality (PSQI) and maternal mood (EPDS) scores. This analysis revealed that $r24$ of activity was significantly related with cortisol amplitude ($r = 0.33$; $p = 0.01$) and negatively related to PSQI score ($r = -0.17$; $p = 0.05$). There was no relationship to mood scores.

## Discussion

Analysis of 24 hr rest-activity and saliva cortisol rhythms across the second and third trimester of gestation and in the early postpartum period showed that rhythms changed as women transitioned through these unique metabolic-physiological states. In general, circadian rhythms of activity progressively diminished across gestation and into the postpartum. More robust activity rhythms associated with more robust cortisol rhythms. Moreover, potential relationships among rest-activity, sleep and cortisol rhythms with maternal-fetal health indicators were revealed. The final model across all study time points indicated 11% of cortisol amplitude was explained by the quality of circadian rest-activity rhythm, as indicated by the autocorrelation coefficient ($r24$), and 3% of the cortisol amplitude was explained by the subjective sleep (PSQI) score. Autocorrelation coefficient was also positively associated with gestational age of infant at birth, suggesting that more irregular sleep-activity rhythms may be associated with earlier term births. To our knowledge, this pilot study is one of the first investigations reporting on objectively measured rest-activity rhythms and the longitudinal relationship to salivary cortisol rhythms and pregnancy disorders.

### Circadian rhythms become attenuated as pregnancy progresses and with the onset of the early postpartum period

Physiologic-metabolic adaptations occur as females transition through reproductive states, with maternal circadian rhythms of behavior and physiology changing throughout pregnancy and lactation [23–30]. Previous studies reported overall activity levels of pregnant women were decreased and onset of activity shifted to an earlier time relative to non-pregnant states [31]. We found no difference in time of waking or start of sleep time between gestational weeks 22 and 32 in this population of women [18], but there was a shift in time of peak of activity to nearly an hour earlier in the third trimester relative to second trimester. Mean activity (mesor) of women in our population also decreased as pregnancy progressed from gestational weeks 22 to 32. There was a further decline in activity in the early postpartum period, and time of peak activity shifted back to the later time recorded for gestational week 22. The quality of rest-activity rhythms, as indicated by $r24$, decreased between gestational weeks 22 and 32 and then again between the third trimester and the first week postpartum, which is consistent with findings of others [32].

We found that cortisol mesor, AUC and amplitude were not different between the gestational time points, however, mesor, AUC and amplitude significantly decreased between the

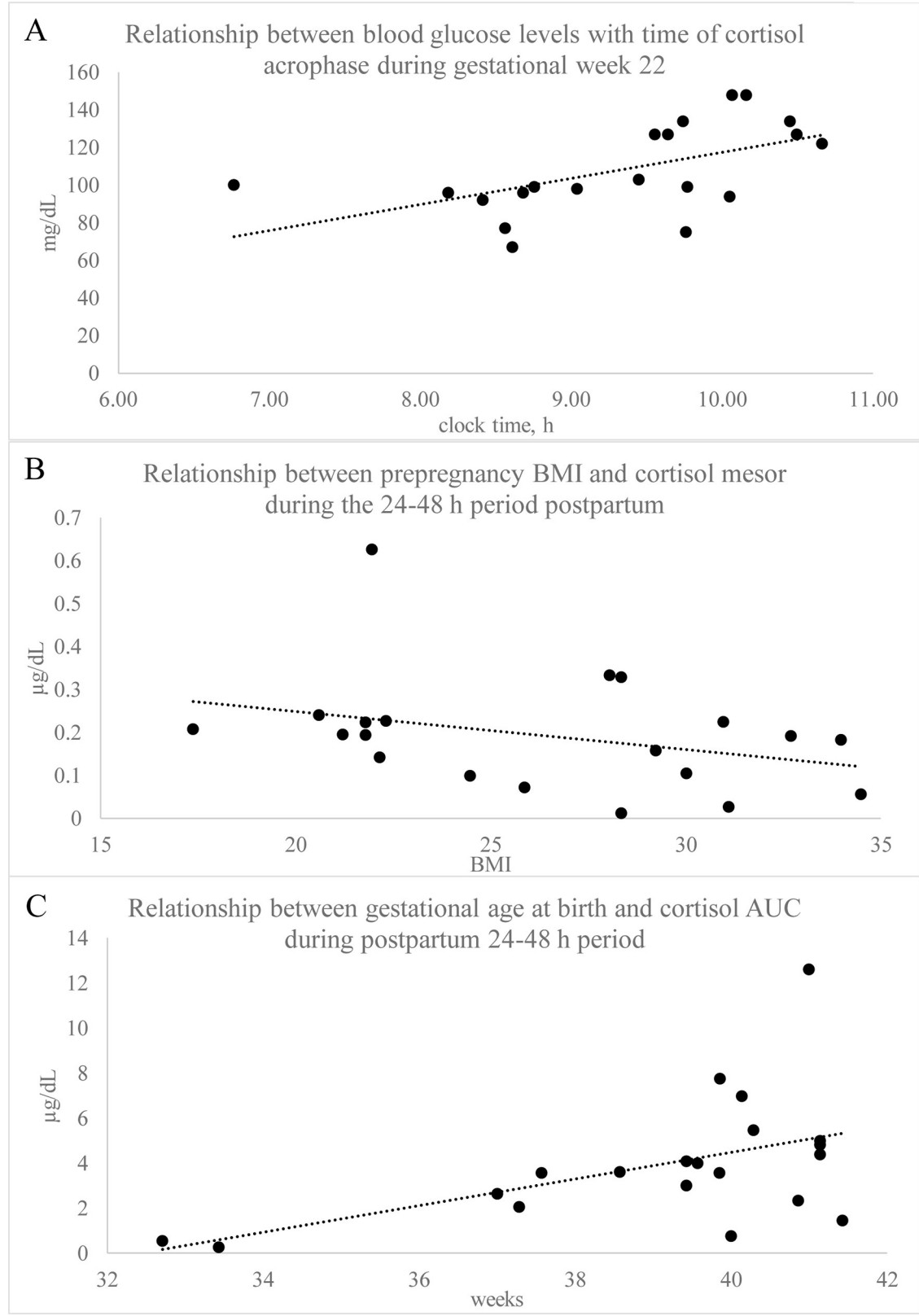

**Fig 5. Relationship of saliva cortisol circadian rhythm variables to maternal-fetal health inidicators.** Relationship between blood glucose levels with time of cortisol acrophase during gestational week 22, $r = 0.58$, $p = 0.006$ (A); Relationship between prepregnancy

BMI and cortisol mesor during the 24–48 h period postpartum, $r = -0.47$, $p = 0.03$ (B); Relationship between gestational age at birth and cortisol AUC during postpartum 24–48 h period, $r = 0.50$, $p = 0.02$.

third trimester and 24–48 h postpartum. Comparison of cortisol concentrations measured in our study with those previously reported, found highly similar saliva concentrations during gestational week 32 and the postpartum sampling period [33]. The magnitude of drop of cortisol levels from gestational week 32 to the early postpartum was also similar between our studies [33]. However, levels of salivary cortisol were higher during gestational week 22 in our study than those reported previously by this same group [33].

During gestational week 22 the timing of cortisol acrophase was significantly related to the timing of peak activity. However, the significant association was lost by the third trimester. We speculate that this may be because circadian rhythms are masked by other dominant physiological changes in women that are underway in the third trimester. Alternatively, a general dampening in circadian rhythms associated with later stages of gestation may have caused loss of association and reflect maternal physiological adaptations needed to support this reproductive state. In particular, circadian rhythms of rest-activity dampened in women between gestational week 22 and 32. Although cortisol mesor and amplitude were not different between gestational time points in our study, it is generally accepted that rhythms of cortisol become attenuated as pregnancy progresses. Attenuation of circadian rhythms of behavior and physiology have also been reported for patterns of food intake in pregnant and lactating rodents [30], and circadian rhythms of gene expression [34]. More studies are needed to understand the interaction between circadian-metabolic systems in pregnant and lactating females to understand the significance of changes.

## Less robust activity rhythms and elevated cortisol levels are associated with poorer maternal-fetal health indicators

Despite the general dampening of circadian rhythms as pregnancy progresses, analysis of the association of rest-activity circadian rhythm parameters with health indicators, support the importance of maintaining circadian rhythms for maternal-fetal health. In particular, more regular rest-activity rhythms during gestational week 32, as reflected in higher $r24$, were associated with greater gestational age of infant at birth. Moreover, women diagnosed with gestational related disease or BMI greater than 25 had lower amplitudes of activity and poorer fitting of rest-activity patterns to 24 hr rhythms. Circadian and metabolic systems are intimately linked and disruption of circadian rhythms is often associated with development of metabolic syndrome, obesity, and diabetes [35–38], and the association of higher BMI with poorer rest-activity rhythms is consistent with findings of studies of nonpregnant women [39, 40].

Activity of the hypothalamic-pituitary-adrenal (HPA) axis is dysregulated in obese individuals [41]. Dysregulation of the HPA axis in non-pregnant obese women is evident in attenuation of the cortisol circadian rhythm, higher basal levels and alterations in stress response [41]. Studies of others found that during the third trimester, women with prepregnancy obesity had elevated evening cortisol versus normal weight women [42]. We did not find this in our study, but interestingly found that in the 24–48 h period immediately postpartum there was a negative relationship between cortisol mesor and amplitude with prepregnancy body mass index. In the third trimester, women diagnosed with gestational-related disease showed a trend for higher cortisol levels, but in the postpartum 24–48 hr period cortisol levels were significantly lower in women with disease than women without. The trend for higher cortisol levels

prepartum in women diagnosed with gestational disease is consistent with the relationship of hypercortisolism with gestational diabetes [43, 44], as well as gestational hypertension and pre-eclampsia [45]. We speculate that the significantly lower levels of free cortisol in the postpartum period in women with higher BMI or diagnosis of gestational related disease reflects a potential refractory period in the HPA axis related to these pathological states. We plan to investigate this further in a larger study population.

The time of cortisol acrophase was positively related to blood glucose levels during gestational week 22. Thus, when cortisol level peaked later in the day, it tended to associate with elevated blood glucose (hyperglycemia). Cortisol regulates blood glucose, with cortisol increasing circulating blood glucose, and consistent with this, is the similar phases of circadian rhythms of cortisol and glucose [46]. Glucose tolerance and clearance also vary by time of day in pregnant women [47, 48]. The association between later cortisol acrophase and higher blood glucose maybe indicative of dysregulation of the HPA axis in individuals with hyperglycemia [44]. Moreover, during the second trimester the minutes of wake after sleep onset (WASO) was positively related to cortisol levels, whereas sleep efficiency was negatively related to cortisol levels. These relationships suggest that poorer sleep is associated with higher cortisol levels during pregnancy. Meta-analysis found a negative association between maternal saliva cortisol and infant birth weight [49]. Women with Cushing's disease or Cushing's syndrome, which is characterized by hypercortisolism, during pregnancy have an increased risk for small-for-gestational-age babies and adverse pregnancy outcomes [50, 51]. Studies of sheep found that increasing maternal cortisol concentrations slowed fetal growth rate and was associated with alterations in glucose homeostasis [52]. Together findings support a potential for a mechanistic link between sleep disruption and elevated cortisol, which alters glucose homeostasis and has associated negative effects on maternal-offspring health and thus warrant further investigation.

## Considerations of study population and study limitations

Women recruited for the study were primarily of lower socioeconomic status. Pregnant women of lower socioeconomic status were found to have lower cortisol awakening responses and less of a change in cortisol levels across a day [53], indicating a general dampening of cortisol rhythms and responses in this population. Across all study time points $r24$ was significantly related with cortisol amplitude, which suggests that rest-activity behaviors affect the quality of cortisol rhythms. Our previous study of this population, found that women who worked daytime shifts had significantly better self-assessed sleep quality than those who worked evening-night or rotating shifts [19]. The subset of the population that was unemployed had a wide variability in sleep quality scores, resulting in no difference in sleep quality between group and day-time or shift/night workers. In this study, we found that across all time points the more-robust activity rhythms were associated with better self-assessed sleep scores (PSQI score). Together, these findings support the need for development of studies designed to test if interventions that target maintaining regular daily activity behaviors during pregnancy regulate circadian rhythms of physiology to better insure maternal-fetal health, particularly among women of lower socioeconomic status.

The small sample size of this pilot study limited the ability to establish strong relationships between rest-activity rhythms and maternal-fetal health. Establishing a relationship between cortisol circadian variables and maternal-fetal health was even more limited by sample size. Moreover, the study relied on participants self-reporting time of saliva sampling instead of using track-cap devices, which electronically record the time of sampling. Studies have found that nonadherence to saliva sampling in ambulatory settings can significantly affect cortisol

variables [54]. Additionally, saliva cortisol concentration was measured using ELISA. Although, studies found that immunoassay analysis of saliva cortisol concentration largely comparable to liquid chromatography tandem mass spectrometry (LC-MS/MS), immunoassay values were consistently greater than LC-MS/MS and the limit of detection was 5 nmol/L (0.18 μg/dL) or more [55]. Despite limitations, these data support the need for further investigations into the interactions among circadian rhythms of cortisol with rest-activity rhythms, sleep metrics and maternal-fetal health.

## Conclusion

Rest-activity rhythms were related to cortisol rhythms in pregnant women, with amplitude of cortisol related to rest-activity circadian rhythm quality. Variables reflecting relative quality of rest-activity circadian rhythms during pregnancy were related to gestational age of infant at birth, and thus may reflect maternal-fetal health. Study findings also support a relationship between cortisol levels, blood glucose, and rest-activity rhythms, and thus call for further investigations to determine if circadian system and sleep disruption affect the metabolic-hormonal systems in a manner that potentially impacts maternal-fetal health.

## Supporting information

**S1 Fig. Study design.**
(DOCX)

**S2 Fig.** Actigraphs of a representative single study participant across study time points and cosine fitted curve (red) during gestational week 22 (A), 32 (B) and postpartum week one (C). Every 30-sec epoch of activity each day was averaged across all recorded days within a subject, and then these data were averaged across subjects to express data within a 24 h period and cosine fitted curve (red) was calculated for gestational week 22 (D), 32 (E) and postpartum week one (F).
(DOCX)

**S3 Fig.** Spaghetti graphs of saliva cortisol concentration of each study participant during gestational week 22 (A), 32 (B) and postpartum week one (C). Cortisol level units are μg/dL.
(DOCX)

## Author Contributions

**Conceptualization:** Theresa Casey, Lingsong Zhang, Karen Plaut, Azza Ahmed.

**Data curation:** Theresa Casey, Shelley Dowden, David M. Haas, Azza Ahmed.

**Formal analysis:** Theresa Casey, Hui Sun, Aridany Suarez-Trujillo, Jennifer Crodian, Lingsong Zhang, Helen J. Burgess, David M. Haas.

**Funding acquisition:** Theresa Casey.

**Investigation:** Theresa Casey, Hui Sun, Jennifer Crodian, Lingsong Zhang, Shelley Dowden, David M. Haas, Azza Ahmed.

**Methodology:** Theresa Casey, Lingsong Zhang, David M. Haas, Azza Ahmed.

**Project administration:** Theresa Casey, Shelley Dowden, David M. Haas.

**Supervision:** Theresa Casey, David M. Haas.

**Writing – original draft:** Theresa Casey.

**Writing – review & editing:** Hui Sun, Aridany Suarez-Trujillo, Jennifer Crodian, Lingsong Zhang, Karen Plaut, Helen J. Burgess, Shelley Dowden, David M. Haas, Azza Ahmed.

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
