## [Decision Letter · Decision Letter 0]

6 Jan 2020

PONE-D-19-29597

Longitudinal relationship between rest-activity and cortisol circadian rhythms during pregnancy with maternal-fetal health: A pilot study.

PLOS ONE

Dear Dr. Casey,

Thank you for submitting your manuscript to PLOS ONE. After careful consideration, we feel that it has merit but does not fully meet PLOS ONE’s publication criteria as it currently stands. Therefore, we invite you to submit a revised version of the manuscript that addresses the points raised during the review process.

The reviewers have been rather divides on this manuscript, one recommending "rejection". However, I find that the paper has some merits, accordingly I allow you to improved the paper taking the criticism into account. In addition to the issues raised by the reviewers please, clarify how the gestational age was determined and provide information on pre-analytic coefficient of variation.

We would appreciate receiving your revised manuscript by Feb 20 2020 11:59PM. To enhance the reproducibility of your results, we recommend that if applicable you deposit your laboratory protocols in protocols.io, where a protocol can be assigned its own identifier (DOI) such that it can be cited independently in the future. For instructions see: http://journals.plos.org/plosone/s/submission-guidelines#loc-laboratory-protocols

We look forward to receiving your revised manuscript.

Kind regards,

Pal Bela Szecsi, M.D. D.M.Sci.

Academic Editor

PLOS ONE

Journal Requirements:

2.  Please provide additional details regarding participant consent.

Specifically, please add information on whether consent for using newborn data was sought from the parents. In the ethics statement in the Methods and online submission information, please ensure that you have specified (i) whether consent was suitably informed and (ii) what type you obtained (for instance, written or verbal).

If your study included minors under age 18, state whether you obtained consent from parents or guardians.

If the need for consent was waived by the ethics committee, please include this information.

Reviewers' comments:

Reviewer's Responses to Questions

**Comments to the Author**

1. Is the manuscript technically sound, and do the data support the conclusions?

Reviewer #1: Partly

Reviewer #2: Partly

2. Has the statistical analysis been performed appropriately and rigorously? 

Reviewer #1: No

Reviewer #2: Yes

3. Have the authors made all data underlying the findings in their manuscript fully available?

Reviewer #1: No

Reviewer #2: Yes

4. Is the manuscript presented in an intelligible fashion and written in standard English?

Reviewer #1: Yes

Reviewer #2: Yes

5. Review Comments to the Author

Reviewer #1: PONE-D-29597

The HPA-axis is influenced by a wide variety of factors in health and disease. The present manuscript has the title “Longitudinal relationship between rest-activity and cortisol circadian rhythms during pregnancy with maternal-health: A pilot study.

1. The title, like the manuscript itself, struggles with stating unequivocally what the focused aim and results of the study are. The question is whether it the relationship between rest-activity and cortisol circadian rhythms in pregnancy or cortisol and maternal health or both?

2. A shorter and more pointed title, e.g. “The effects of maternal health and rest-activity on salivary cortisol concentrations in pregnancy” gives a clearer picture.

3. The rest-activity registration and advanced analysis of the cortisol diurnal variation play a crucial role when describing the results of the study, but the introduction and discussion parts of the manuscript contain voluminous references to cortisol concentrations in other contexts. The authors are advised to shorten the manuscript by at least a third by removing text that only marginally supports the aim and results of the study.

4. The authors refer to the two following publications for further information about the sample of pregnant women from the population: Ahmed AH, Hui S, Crodian J, Plaut K, Haas D, Zhang LS, et al. Relationship Between Sleep Quality, Depression Symptoms, and Blood Glucose in Pregnant Women. Western J Nurs Res. 2019;41(9):1222-40. And Casey T, Sun H, Burgess HJ, Crodian J, Dowden S, Cummings S, et al. Delayed Lactogenesis II is Associated With Lower Sleep Efficiency and Greater Variation in Nightly Sleep Duration in the Third Trimester. J Hum Lact. 2019;35(4):713-24. In the opinion of the present reviewer it is absolutely crucial to include the combined comprehensive information provided in the two manuscripts in the description of the population and the sample from the population used in the study. The concept “convenience sample” should be used to use an established term to describe the sampling. It is also crucial to describe the dropout rate and discuss the risks of confounding caused by the selection procedure and the dropout.

5. The research design used has possibly been frowned on by earlier reviewers and editors. My take on the matter is different. In order to study the functions of the HPA-axis in humans, it may be a substantial advantage to study the HPA axis under stress as in the present study. Substantial stress is obviously more likely to be found in a group of women from a disadvantageous population of pregnant women. My suggestion is therefore that the authors see the group of pregnant women they have studied primarily as an advantage rather than a problem due to dropouts etc. They may therefore even consider stating this flat out in the title, e.g. “The effects of maternal health and rest-activity on salivary cortisol concentrations in pregnancy in women from a disadvantaged population” and clearly point out the advantages of their study design which also evidently results in the dropouts etc. inevitably occurring.

6. The description of the demographics of the study population on page 11 lines 201-211 should be given in the Material and Methods section of the manuscript as part of the research design and not in the results section of the manuscript as unexpected challenges in the results of the study.

7. The authors are encouraged to make clearer in the manuscript which effect they consider the rest-activity of the subjects have on the cortisol circadian rhythmicity.

8. The two last lines on page 18 report the Pearson r correlation coefficient. R=0.33 means that 0.33^2=0.1089 meaning that 11% of the cortisol amplitude is explained by the r24 and that 0.17^2=0.0289 meaning that 3% of the cortisol amplitude is explained by the PSQI score. In the eyes of the present reviewer this type of information is even more informative than the r values and probability statements themselves.

9. The last sentence of the abstract (page 2, lines 38-40) should be deleted since it only states the obvious.

10. Page 3, line 64 reads “pregnancy total plasma cortisol”. The concept “total plasma cortisol rises progressively” is only logical if “total concentration of cortisol rises progressively” is meant. Since the authors measure the concentrations of cortisol in saliva as reflection of the free fraction of cortisol in plasma the concept concentration seems reasonable.

11. Page 4, line 67-68 states “which is the biologically active component in serum”. The authors will agree that serum is a component of blood which only exists if whole blood is allowed to coagulate fully and is subsequently centrifuged. Serum never exists in vivo in a patient. Line 68 should therefore read “component in plasma (14)”.

12. Page 8, line 147 should read “All samples were run in duplicate and the mean value of the duplicate results used.”

13. The authors should detail the method(s) and probability criteria used for outlier analysis

Reviewer #2: Casey et al investigate the relationship between rest activity rhythms and salivary cortisol rhythms and how they are related to maternal fetal health indicators. Various relationships are found between variables. The paper is written well but various issues need addressing.

1. It is not clear what the primary objective of the study was and various findings are presented with no focus on the most important. What is the primary aim of the study? What is the main question?

2. It would have been useful to have a control group of individuals that are not pregnant to assess how different from normal physiology the changes we are observing are. Maybe comparison to model data in the literature could be useful.

3. Health status was one of the reasons for a large number of withdrawals. Could important data have been lost and could this have influenced the final outcomes as a number of health status variables were being assessed?

4. Are the postpartum changes for rest activity and cortisol clinically significant? Is the difference to gestational levels relevant and why. this must be discussed

5. Saliva cortisol is measured by immunoassay not LC-MS/MS. This is a limiting factor of the study and should be discussed.

6. The units for AUC should have a time component.

7. How does one explain the lower cortisol mesor postpartum in those with gestational disease than those without gestational disease?

8.Figure 3 could be improved by showing a spaghetti chart for all individual cortisol rhythms. Figure 3D does not give that much useful information and one could draw the three mean cosinors for each time point

6. PLOS authors have the option to publish the peer review history of their article (what does this mean?). If published, this will include your full peer review and any attached files.

Reviewer #1: Yes: Elvar Theodorsson

Reviewer #2: No

---

## [Author Response · Author response to Decision Letter 0]

31 Jan 2020

Responses to Editor’s and Reviewers’ Comments 

Editor.

done 

2. Specifically, please add information on whether consent for using newborn data was sought from the parents. In the ethics statement in the Methods and online submission information, please ensure that you have specified (i) whether consent was suitably informed and (ii) what type you obtained (for instance, written or verbal).

Information was added.

Title was edited in both online submission and title page of manuscript.

Both reviewers commented that data needed to be made freely accessible, and so were deposited here: https://purr.purdue.edu/publications/3376

Reviewer #1: PONE-D-29597

The HPA-axis is influenced by a wide variety of factors in health and disease. The present manuscript has the title “Longitudinal relationship between rest-activity and cortisol circadian rhythms during pregnancy with maternal-health: A pilot study.

1. The title, like the manuscript itself, struggles with stating unequivocally what the focused aim and results of the study are. The question is whether it the relationship between rest-activity and cortisol circadian rhythms in pregnancy or cortisol and maternal health or both?

RE: In order to better frame, and justify the pilot study we rewrote the opening of the introduction and stream-lined the content. The objectives of the study were also rewritten, and hypothesis statement added. 

The abstract was also edited to better highlight importance of study and relevant findings.

2. A shorter and more pointed title, e.g. “The effects of maternal health and rest-activity on salivary cortisol concentrations in pregnancy” gives a clearer picture.

The title was changed to make more pointed.

3. The rest-activity registration and advanced analysis of the cortisol diurnal variation play a crucial role when describing the results of the study, but the introduction and discussion parts of the manuscript contain voluminous references to cortisol concentrations in other contexts. The authors are advised to shorten the manuscript by at least a third by removing text that only marginally supports the aim and results of the study.

Introduction was rewritten and streamlined to only pertinent information as recommended.

4. The authors refer to the two following publications for further information about the sample of pregnant women from the population: Ahmed AH, Hui S, Crodian J, Plaut K, Haas D, Zhang LS, et al. Relationship Between Sleep Quality, Depression Symptoms, and Blood Glucose in Pregnant Women. Western J Nurs Res. 2019;41(9):1222-40. And Casey T, Sun H, Burgess HJ, Crodian J, Dowden S, Cummings S, et al. Delayed Lactogenesis II is Associated With Lower Sleep Efficiency and Greater Variation in Nightly Sleep Duration in the Third Trimester. J Hum Lact. 2019;35(4):713-24. In the opinion of the present reviewer it is absolutely crucial to include the combined comprehensive information provided in the two manuscripts in the description of the population and the sample from the population used in the study. The concept “convenience sample” should be used to use an established term to describe the sampling. It is also crucial to describe the dropout rate and discuss the risks of confounding caused by the selection procedure and the dropout.

The total enrollment under population was added to the methods section as well as other details. ‘Convenience sample’ was added. 

5. The research design used has possibly been frowned on by earlier reviewers and editors. My take on the matter is different. In order to study the functions of the HPA-axis in humans, it may be a substantial advantage to study the HPA axis under stress as in the present study. Substantial stress is obviously more likely to be found in a group of women from a disadvantageous population of pregnant women. My suggestion is therefore that the authors see the group of pregnant women they have studied primarily as an advantage rather than a problem due to dropouts etc. They may therefore even consider stating this flat out in the title, e.g. “The effects of maternal health and rest-activity on salivary cortisol concentrations in pregnancy in women from a disadvantaged population” and clearly point out the advantages of their study design which also evidently results in the dropouts etc. inevitably occurring.

Thank you for this comment. We added the phrase ‘from a disadvantage population’. Because of the pilot nature and design of the study we limited title to claims of ‘relationships’. 

6. The description of the demographics of the study population on page 11 lines 201-211 should be given in the Material and Methods section of the manuscript as part of the research design and not in the results section of the manuscript as unexpected challenges in the results of the study.

Recruitment information was moved to the Methods section. 

7. The authors are encouraged to make clearer in the manuscript which effect they consider the rest-activity of the subjects have on the cortisol circadian rhythmicity.

We removed descriptive changes in rest-activity and cortisol rhythms from abstract, to help highlight the significant relationships between these rhythms, and maternal-fetal health.

8. The two last lines on page 18 report the Pearson r correlation coefficient. R=0.33 means that 0.33^2=0.1089 meaning that 11% of the cortisol amplitude is explained by the r24 and that 0.17^2=0.0289 meaning that 3% of the cortisol amplitude is explained by the PSQI score. In the eyes of the present reviewer this type of information is even more informative than the r values and probability statements themselves.

These findings were highlighted in the opening of the discussion and conclusion. 

9. The last sentence of the abstract (page 2, lines 38-40) should be deleted since it only states the obvious.

Deleted.

10. Page 3, line 64 reads “pregnancy total plasma cortisol”. The concept “total plasma cortisol rises progressively” is only logical if “total concentration of cortisol rises progressively” is meant. Since the authors measure the concentrations of cortisol in saliva as reflection of the free fraction of cortisol in plasma the concept concentration seems reasonable.

Changed as suggested. 

11. Page 4, line 67-68 states “which is the biologically active component in serum”. The authors will agree that serum is a component of blood which only exists if whole blood is allowed to coagulate fully and is subsequently centrifuged. Serum never exists in vivo in a patient. Line 68 should therefore read “component in plasma (14)”.

Fixed.

12. Page 8, line 147 should read “All samples were run in duplicate and the mean value of the duplicate results used.”

Fixed.

13. The authors should detail the method(s) and probability criteria used for outlier analysis

No outlier analysis was implemented in our statistical analysis because the sample size was so small. In the boxplots, the potential outlier (with star) is calculated by 5-number summary method where any value > Q3 + 1.5(Q3-Q1) or < Q1- 1.5(Q3-Q1) is considered a potential outlier (Q1 and Q3 here and 1st and 3rd quartile respectively). Due to the highly skewed data with potential outliers, we use rank based methods (like Spearman analysis) without assuming normality for the data.

Reviewer #2: Casey et al investigate the relationship between rest activity rhythms and salivary cortisol rhythms and how they are related to maternal fetal health indicators. Various relationships are found between variables. The paper is written well but various issues need addressing.

1. It is not clear what the primary objective of the study was and various findings are presented with no focus on the most important. What is the primary aim of the study? What is the main question?

Reviewer #1 had similar comments, we addressed these weaknesses by rewriting the first lines of abstract and first and last paragraphs of introduction. We also streamlined content of Introduction and Discussion.

2. It would have been useful to have a control group of individuals that are not pregnant to assess how different from normal physiology the changes we are observing are. Maybe comparison to model data in the literature could be useful.

Previous studies were conducted by others to determine differences in activity and cortisol rhythms between pregnant and non-pregnant women. This work is referred to in our manuscript. We also refer to data that measure and assessed rest-activity and relation to hormonal rhythms in non-pregnant states. Our aim was to determine if there is a relationship between rest-activity and cortisol rhythms during gestation and between these rhythms and maternal-fetal health indicators.

3. Health status was one of the reasons for a large number of withdrawals. Could important data have been lost and could this have influenced the final outcomes as a number of health status variables were being assessed?

We apologize for mis-stating in original. Reasons for half withdrawals were: loss of fetus, moving out of area or other life events were reasons for half of study withdrawals

4. Are the postpartum changes for rest activity and cortisol clinically significant? Is the difference to gestational levels relevant and why. this must be discussed

Very few studies have been conducted to measure these variables, and thus the need for the type of study we conducted. At this time, we do not know if there is any clinical significance. We feel that we are limited in reporting or discussing anything beyond what we did because of the limited sample size. Thus, it remains an observation that in this sample of women, levels of cortisol were significantly lower in women with BMI >25 and who were diagnosed with gestationally-related disease. Additionally, we add that this may potentially be indicative of a refractory period in functioning of HPA, and that further research is needed. 

5. Saliva cortisol is measured by immunoassay not LC-MS/MS. This is a limiting factor of the study and should be discussed.

Added to limitations

6. The units for AUC should have a time component.

Added throughout, per 24 hr

7. How does one explain the lower cortisol mesor postpartum in those with gestational disease than those without gestational disease?

At this time we have no explanation, it is just an observation, that we speculate may be indicative of a refractory period in the HPA axis in this population.

8.Figure 3 could be improved by showing a spaghetti chart for all individual cortisol rhythms. Figure 3D does not give that much useful information and one could draw the three mean cosinors for each time point

Spaghetti charts were added as a supplemental figure. The lines in figure 3D, represent the cosinor curve calculated across individuals from data collected at each time period, and used to generate mean mesor, amplitude and AUC that is reported. We decided to keep, as it can be a reference/visualization tool for the reader.

---

## [Decision Letter · Decision Letter 1]

11 Feb 2020

Pregnancy rest-activity patterns are related to salivary cortisol rhythms and maternal-fetal health indicators in women from a disadvantaged population

PONE-D-19-29597R1

Dear Dr. Casey,

We are pleased to inform you that your manuscript has been judged scientifically suitable for publication and will be formally accepted for publication once it complies with all outstanding technical requirements.

With kind regards,

Pal Bela Szecsi, M.D. D.M.Sci.

Academic Editor

PLOS ONE

Additional Editor Comments (optional):

Reviewers' comments:

Reviewer's Responses to Questions

**Comments to the Author**

1. If the authors have adequately addressed your comments raised in a previous round of review and you feel that this manuscript is now acceptable for publication, you may indicate that here to bypass the “Comments to the Author” section, enter your conflict of interest statement in the “Confidential to Editor” section, and submit your "Accept" recommendation.

Reviewer #1: All comments have been addressed

Reviewer #2: All comments have been addressed

2. Is the manuscript technically sound, and do the data support the conclusions?

Reviewer #1: Yes

Reviewer #2: Yes

3. Has the statistical analysis been performed appropriately and rigorously? 

Reviewer #1: Yes

Reviewer #2: Yes

4. Have the authors made all data underlying the findings in their manuscript fully available?

Reviewer #1: Yes

Reviewer #2: Yes

5. Is the manuscript presented in an intelligible fashion and written in standard English?

Reviewer #1: Yes

Reviewer #2: Yes

6. Review Comments to the Author

Reviewer #1: The suggestions for improvements of the reviewer have been addressed and the manuscript is substantially improved.

Reviewer #2: Thanks for changes and for responding to all queries. The article has improved considerably and should be good for publication

7. PLOS authors have the option to publish the peer review history of their article (what does this mean?). If published, this will include your full peer review and any attached files.

Reviewer #1: Yes: Elvar Theodorsson

Reviewer #2: No

---

## [Editor Report · Acceptance letter]

14 Feb 2020

PONE-D-19-29597R1 

Pregnancy rest-activity patterns are related to salivary cortisol rhythms and maternal-fetal health indicators in women from a disadvantaged population 

Dear Dr. Casey:

I am pleased to inform you that your manuscript has been deemed suitable for publication in PLOS ONE. Congratulations! Your manuscript is now with our production department. 

With kind regards,

on behalf of

Dr. Pal Bela Szecsi 

Academic Editor

PLOS ONE